# Causal implicatures from correlational statements

**Samuel J. Gershman** [ID] *, **Tomer D. Ullman** *

Department of Psychology, Harvard University, Cambridge, MA, United States of America

* gershman@fas.harvard.edu (SJG); tullman@fas.harvard.edu (TDU)

**Data Availability Statement:** All data are available at https://osf.io/u34ex/?view_only=1ffdf6729af149d6b14f7e32692c0334.

**Funding:** This work was supported by the Center for Brains, Minds and Machines (CBMM), funded by NSF STC award CCF1231216. There was no

## Abstract

Correlation does not imply causation, but this does not necessarily stop people from drawing causal inferences from correlational statements. We show that people do in fact infer causality from statements of association, under minimal conditions. In Study 1, participants interpreted statements of the form "X is associated with Y" to imply that Y causes X. In Studies 2 and 3, participants interpreted statements of the form "X is associated with an increased risk of Y" to imply that X causes Y. Thus, even the most orthodox correlational language can give rise to causal inferences.

## Introduction

Modern scientists are carefully trained to avoid conflating causation and correlation when describing research results. A correlation between two variables may reflect the causal effect of one variable on the other, or the causal effect of another variable on both. For this reason, observational studies report findings using seemingly non-causal stock phrases: variables are "associated" or "linked" with one another. However, it is possible that readers of these correlational statements may interpret them as causal.

Much work has demonstrated that people draw pragmatic inferences from ambiguous or incomplete linguistic utterances [1–3]. In particular, studies of "implicit causality" in language understanding have shown that people draw inferences about ambiguous causal roles from verbs [4–6]. For example, people infer that "she" refers to the daughter in the following sentence: "The mother punished her daughter because she admitted her guilt" [4]. In contrast, people infer that "she" refers to the mother in the following sentence: "The mother punished her daughter because she discovered her guilt." The verbs "admitted" and "discovered" induce different causal role assignments. Similar results have been reported for sentence completion tasks, where the referent of the completed sentence picks out different nouns depending on the verb.

One influential view of implicit causality is that it reflects inferences about event structure [5], rather than reflecting linguistic structure (though see [6]). Supporting this view is evidence that causal role assignments are sensitive to general knowledge and social context, such as the gender and typicality of the agents/patients in the sentence. More broadly, the literature on implicit causality suggests that language is a rich source of causal information.

additional external funding received for this study. The funders had no role in study design, data collection and analysis, decision to publish, or preparation of the manuscript.

**Competing interests:** The authors have declared that no competing interests exist.

For our purposes, an important limitation of the implicit causality concept is that it takes as given some causal background knowledge (e.g., that mothers punish daughters when they admit guilt) and asks how people use this knowledge to make inferences about linguistic referents. Our goal in this paper is to flip this around: what happens when people know the referents but not the causal background knowledge? This situation is commonly encountered when people are reading newspaper headlines about scientific discoveries: if scientists report that eating ice cream increases the risk for cancer, a natural question is whether ice cream causes cancer. Careful scientists and journalists can expunge causal language from correlation studies, but can they expunge causal representations from the mental models of readers? To answer this question, we undertook a series of studies that assess what kinds of causal inferences people draw from correlational statements.

## Results

We conducted three studies to examine the inference of causality from association statements. The results of Studies 1 and 2 are summarized in Fig 1, and the results of Study 3 are shown in Fig 2. In Study 1, participants were presented with statements of the form "X is associated with Y" and asked to judge whether X caused Y, or Y caused X. In Study 2, participants were presented with statements of the form "X is associated with an increased probability of Y", and asked to make the similar causal judgments. In both studies, we ran versions with nonsense names designed to sound similar to medical terminology (e.g., "Themaglin" or "Pneuben") or arbitrary letter symbols (see Methods for details).

We analyzed the data from Study 1 as follows. If participants chose the first variable as causing the second variable after being presented with the sentence "[X] is associated with [Y]", this was coded as 1, and if they chose the second variable as causing the first, this was coded as 0. All responses within a study were averaged together (288 responses total in the 'nonsense names' condition, 291 responses total in the 'symbols' condition). If people were responding randomly, we would expect the average value to be 0.5. Note that a strategy such as 'just pick the first answer' is negated by the randomization of the variable order in both questions and answers. However, the mean response for both studies was significantly below chance, by a two-sided proportion z-test using Holm-Bonferroni correction for multiple comparisons ('nonsense names' condition: $M = 0.23$, $Z = -11.14$, $SE = 0.025$, $p < 10^{-28}$; 'symbols' condition: $M = 0.38$, $SE = 0.029$, $Z = -4.29$, $p < 10^{-4}$).

These results suggest that when given a simple association statement between two variables, participants inferred that the second variable causes the first. Put plainly, when presented with a sentence such as 'Themaglin is associated with Pneuben', or 'X is associated with Y', participants took this to imply 'Pneuben causes Themaglin', and 'Y causes X'.

The analysis of data from Study 2 followed that of Study 1. If participants chose the first variable as causing the second variable after being presented with the sentence "[X] is associated with [RELATIONSHIP] [Y]", this was coded as 1, otherwise the response was coded as 0. All responses within each relationship and within each study were averaged together. Again, if people were responding randomly, we would expect the average value to be 0.5.

We found that the addition of context drives all responses to be significantly above chance, by a two-sided proportion z-test using Holm-Bonferroni correction for multiple comparisons ('nonsense names' condition: $M_{risk\ increase} = 0.87$, $SE = 0.03$, $Z = 10.58$, $p < 10^{-25}$, $M_{risk\ decrease} = 0.94$, $SE = 0.02$, $Z = 17.91$, $p < 10^{-71}$, $M_{probability\ increase} = 0.88$, $SE = 0.03$, $Z = 11.25$, $p < 10^{-28}$, $M_{probability\ decrease} = 0.89$, $SE = 0.03$, $Z = 12.00$, $p < 10^{-32}$; 'symbols' condition: $M_{risk\ increase} = 0.84$, $SE = 0.04$, $Z = 9.14$, $p < 10^{-19}$, $M_{risk\ decrease} = 0.87$, $SE = 0.03$, $Z = 10.96$, $p < 10^{-27}$, $M_{probability\ increase} = 0.86$, $SE = 0.04$, $Z = 10.30$, $p < 10^{-24}$, $M_{probability\ decrease} = 0.82$, $SE = 0.04$, $Z = 8.16$,

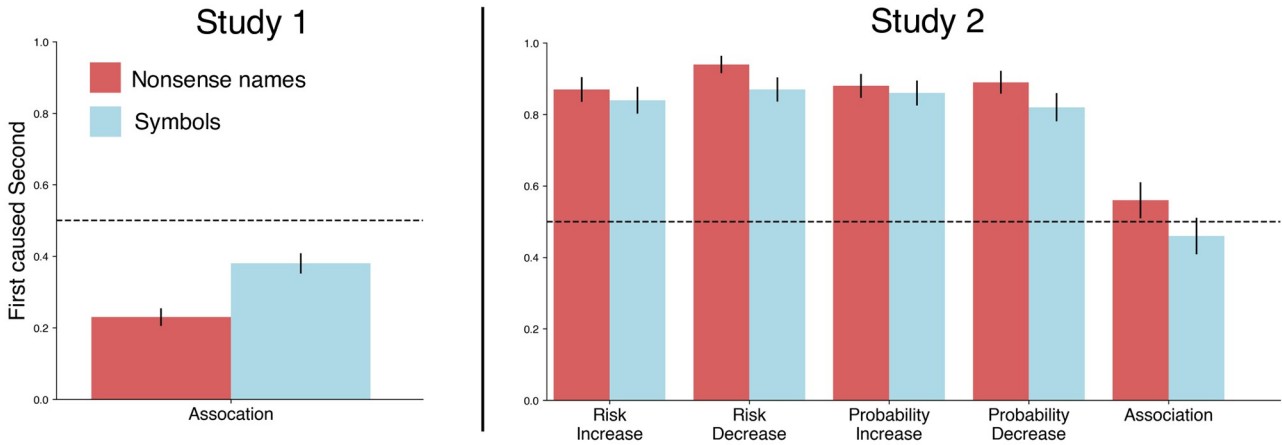

**Fig 1. Summary of results for Study 1 (left) and Study 2 (right).** Participants were given statements such as *'Themaglin is associated with Pneuben'* (Study 1) or *'Denoden is associated with an increased probability of Flembers'* (Study 2). A response of 1 indicates participants interpreted the statement to mean the first variable causes the second, 0 that they took it to mean the second variable causes the first, and 0.5 that they answered at random. The figure shows that without context, simple association is taken to imply the second variable causes the first. With minimal context, the association statement is taken to strongly imply the first variable caused the second. Error bars show standard error estimates for a proportion, $\sqrt{p(1-p)/N}$, where $p = M/N$, $M$ is the number 'Yes' responses, and $N$ is the total number of responses.

$p < 10^{-15}$). By contrast, the simple association statements (which do not include any context) were now not significantly different from chance. Although it is not entirely clear why this aspect of the data did not replicate Study 1, we suspect that it may be related to the context of more causally salient statements (risk increase/decrease, probability increase/decrease) which may have attuned participants to scrutinize association statements more carefully.

One concern with Studies 1 and 2 is that participants were forced to choose one causal direction. We addressed this in Study 3 by giving participants the option to report that neither causal direction was preferred. The results indicate that even with the option of reporting no preferred causal direction, people still typically took the association statement to imply that the first variable caused the second (Fig 2). For the pure association statements, the proportion between 'X → Y' and 'Y → X' was roughly twice that of Study 2, which we take to indicate that in Study 2 some participants were using the 'Y → X' statement to stand in for 'neither are directly related'.

All proportions were different from one another within the different context questions. That is, we ran 3 two-sided proportion z-tests within each context (risk increase, risk decrease, probability increase, probability decrease, and simple association), comparing 'X → Y' to 'Y → X', 'X → Y' to 'Neither', and 'Y → X' to 'Neither, for a total of 15 comparisons, and using the Holm-Bonferroni correction for multiple comparisons (note that the pre-registration has this as 7 questions and so 21 comparisons, due to an typographic error on the part of the experimenters in considering the number of questions asked).

All proportions were also significantly different from the chance response of 33%, except for 'X → Y' in the *Association* context, and 'Neither' in the *Probability Decrease* context. Given the overall pattern of responses, we take these latter two to be coincidental.

## Discussion

These results indicate that when given an association statement between two variables with minimal context that indicates a change in the relationship for the second variable, participants inferred a causal relation, such that the first variable causes the second. In other words,

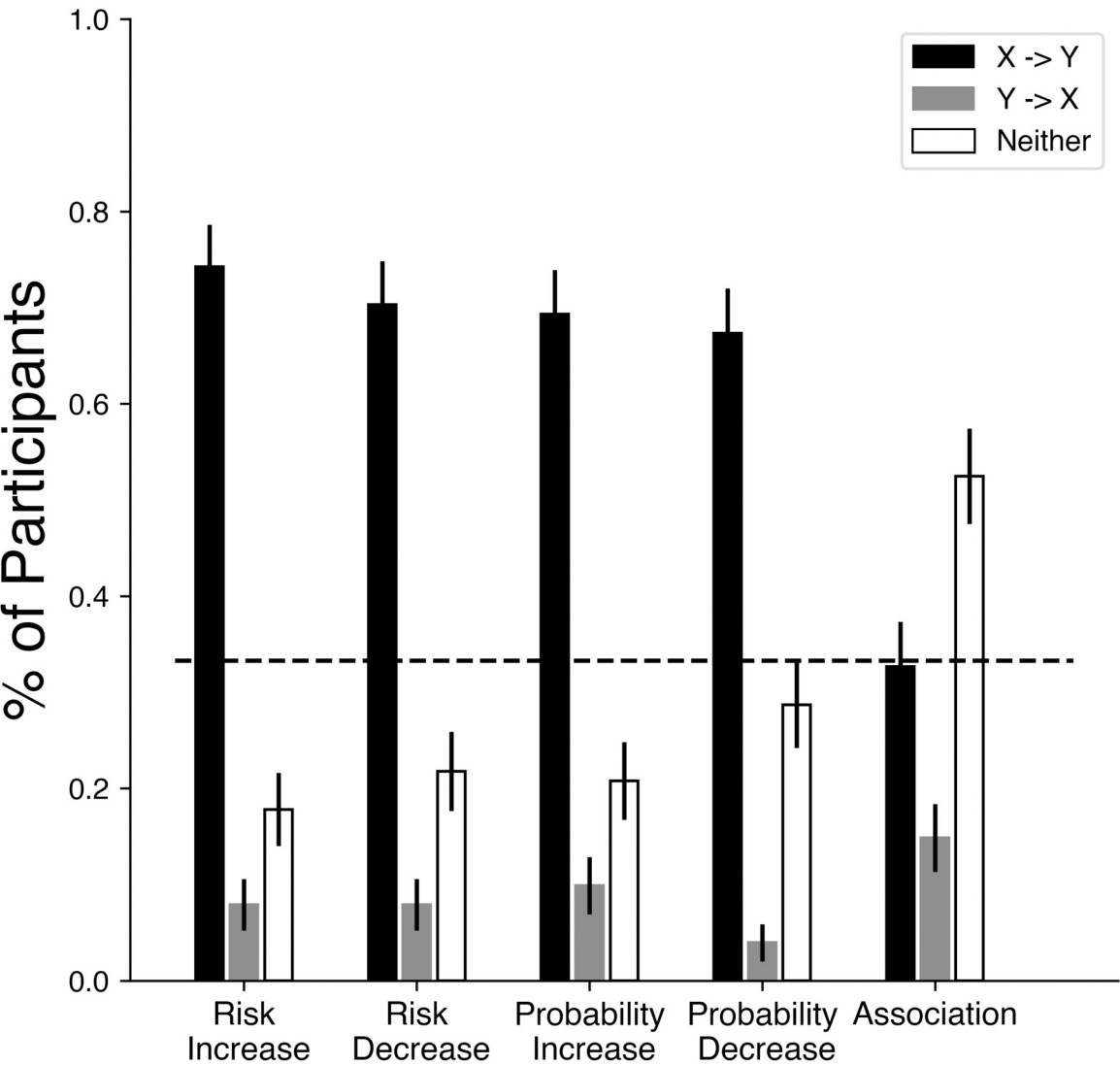

**Fig 2. Summary of results for Study 3.** Participants were given the same statements as in Study 2A, with the addition of a 'Neither' option to report that no causal direction was preferred. The y-axis now indicates the proportion of participants who chose each response. Error bars show standard error estimates for a proportion. The dotted line indicated expected random response levels.

when presented with a sentence such as 'Themaglin is associated with an increased risk of Pneuben', or 'X is associated with an increased risk of Y', participants took this to strongly imply 'Themaglin causes Pneuben', and 'X causes Y'.

While we take the data to show that people draw causal implicatures from correlational statements, a possible concern is that participants were given a forced-choice between two causal relationships, without the option of declaring uncertainty, or rejecting both relationships. But this forced choice was a deliberate feature: because many people are highly drilled in the mantra that correlation does not imply causation, it is likely that when appropriately cued, they will activate the mantra. However, our hypothesis here is that there are implicit causal expectations about linguistic structures that are activated by correlational statements, and these expectations can be made explicit when people are forced to choose between different causal interpretations. If people truly are committed to non-causal interpretations of non-

causal language, then participants should just choose arbitrarily between the different causal interpretations available to them. The fact that we found a large systematic preference argues in favor of our causal implicature hypothesis.

To more directly address the concern that forced choices between causal interpretations might yield artifactual results, we conducted a third study in which participants could choose a 'Neither' option. Strikingly, the results were essentially the same: participants still showed a strong preference for a particular causal direction in all of the experimental conditions apart from the association condition. Thus, it is unlikely that the response format produced a systematic bias.

One potential concern about our findings is that the association condition produced different results in Studies 1 and 2, deviating from random in the former but not in the latter. We speculated that this might be due to some kind of context effect. Study 3 sheds some additional light on this discrepancy, finding that most people endorse a non-causal interpretation of associative statements, although a large minority (47%) still favor a causal interpretation. Within that large minority, we find again a preference for $X \rightarrow Y$ rather than $Y \rightarrow X$. We tentatively propose that many or even most of the people endorsing $Y \rightarrow X$ in Study 2 were doing so as a signal of protest against being asked to assign a causal direction.

Our results cannot unambiguously rule out several alternative hypotheses that hinge on different interpretations of the information we provided to participants. First, it is possible that participants did not distinguish between probabilistic and causal interpretations of the statements. For example, the statement that X increases the risk of Y might also imply that Y increases the risk of X, but the increase for Y given X might still be larger than the increase for X given Y. In this case, there is an asymmetry in the risk pattern, even if there is no causal relationship between X and Y. If participants align their causal preferences with the directional asymmetry, then they might show preferences deviating from random, which we have treated as a signature of causal implicature. In essence, this alternative hypothesis posits either that participants do not fully understand what causality is, or that risk directionality is used as a heuristic for causal implicature. We think it is somewhat unlikely that participants simply do not understand what we are asking when we elicit causal interpretations, given the sophisticated causal reasoning abilities demonstrated in many other studies.

A second alternative hypothesis is that participants interpreted statements of the form "X is associated with increased risk of Y" as "Doing X is associated with increased risk of Y" which would seem to imply a kind of causal intervention. Thus, on this hypothesis apparent causal preferences arise from vagueness in the statements. While we cannot rule out this possibility, we would argue that the vagueness hypothesis is really another version of causal implicature, where participants make a pragmatic interpretation that the speaker intends to communicate a causal relationship.

We have argued that the alignment and vagueness hypotheses may or may not be consistent with a causal implicature hypothesis. More work is required to answer this question.

In summary, certain correlational statements are associated with an increased probability of causal implicature. To be clear, we are not implying that these correlational statements *cause* causal implicature, but rather that they are *correlated with* causal implicature. In other words, correlation does not imply causation, but it does sometimes "imply" causation.

## Methods

The studies were approved by the Harvard Institutional Review Board, protocol no. 19-1861.

Participants were recruited online [7] via the Prolific platform (https://www.prolific.co). Participants were restricted to those located in the USA, and having completed at least 100

prior studies on Prolific, with an acceptance rate of at least 90%. We recruited 100 participants in each study (total of N = 400), following a pilot indicating the expected effect size is such that this number of participants had a power >90%. Participants in any study described here (meaning, each sub-condition) were prohibited from participating in any other study described here.

At the end of each study, participants were asked "Please describe, in a few words, what you were asked to do in this experiment?". We excluded participants who gave nonsensical or non-sequitur answers, such as *'Do the causes'* or *'Opinions'*.

The analyses, experiments, exclusion criteria, and sample sizes were pre-registered, as available here: https://aspredicted.org/blind.php?x=JHK_JR1 and here https://aspredicted.org/blind.php?x=BP6_Z8D.

All data are available at https://osf.io/u34ex/?view_only=1ffdf6729af149d6b14f7e32692c0334.

## Study 1

To examine the basic implicature of the phrasing 'X is associated with Y', without any further context, we designed a simple study in which participants read statements about the existence of an association between nonsense terms such as *Themalgin* and *Pneuben* ('nonsense names' condition), or abstract symbols like *X* and *Y* ('symbols' condition).

**Participants.** We recruited 200 participants, and randomly assigned them evenly to the two conditions ('nonsense names' and 'symbols'). Written/verbal informed consent was obtained from all participants for inclusion in the study. After excluding 4 participants, the mean age of participants in the 'nonsense names' condition was 35.4, and 57 identified as female. After excluding 3 participants in the 'symbols' condition, the mean age of the remaining participants was 34.7, and 66 identified as Female.

**Stimuli.** Participants were informed that they will be asked a few simple questions about the possible relationship between different things, and that the questions are unrelated to one another. Participants then saw 3 questions in succession, each phrased as

*Suppose you read the following piece of information:*

*"[X] is associated with [Y]"*

*Which of the following is more likely?*

Participants were then given a forced choice between the statements *"[X] causes [Y]"* and *"[Y] causes [X]"*.

In the 'nonsense names' condition, [X] and [Y] were replaced with the following nonsense terms: Themaglin, Rebosen, Denoden, Flembers, Agoriv, and Ceflar. In the 'symbols' condition, [X] and [Y] were replaced with the following symbols: X, Y, P, Z, G, and D. The ordering of the nonsense term pairs or symbol pairs within each question, the ordering of the forced choice answers to each question, and the question order of the three questions was randomized.

## Study 2

In Study 2, we provided additional context of the sort that is often found in the scientific literature, as well as the popular press. In particular, we examined the causal implicature of phrases such as 'Ceflar is associated with a lower risk of Agoriv', 'Pneuben is associated with higher probability of Efogen', and so on. The design was similar to Study 1: Participants were given statements about the existence of an association and minimal context about the relationship (risk, probability, increase, decrease) between nonsense terms ('nonsense names' condition), or abstract symbols like *X* and *Y* ('symbols' condition).

**Participants.**   We recruited 200 participants in total, randomly and evenly distributing them to the two conditions ('nonsense names' and 'symbols'). Written/verbal informed consent was obtained from all participants for inclusion in the study. After excluding 3 participants, the mean age of participants in the 'nonsense names' condition was 35.0, and 55 identified as female. After excluding 6 participants in the 'symbols' condition, the mean age of the remaining participants was 34.6, and 70 identified as Female.

**Stimuli.**   Participants were informed that they would be asked a few simple questions about the possible relationship between different things, and that the questions were unrelated to one another. Participants then saw 5 questions in succession, each phrased as:

*Suppose you read the following piece of information:*
*"[X] is associated with [RELATIONSHIP] [Y]"*
*Which of the following is more likely?*

Participants were then given a forced choice between the statements *"[X] causes [CHANGE] [Y]"* and *"[Y] causes [CHANGE] [X]"*.

In the 'nonsense names' condition, [X] and [Y] were replaced with the following nonsense terms: Themaglin, Rebosen, Denoden, Flembers, Agoriv, Ceflar, Pneuben, Efogen, Turilin, and Laurem. In the 'symbols' condition, [X] and [Y] were replaced with the following symbols: T, R, P, E, A, C, X, Y, D, and F. The ordering of the nonsense term pairs or symbol pairs within each question, the ordering of the forced choice answers to each question, and the question order of the five questions was randomized. The [RELATIONSHIP] variable was replaced with: Higher probability, higher risk, lower probability, lower risk, or was left empty. When the [RELATIONSHIP] variable was changed to 'higher' the [CHANGE] relationship was replaced with: 'an increase in'. When the [RELATIONSHIP] variable was changed to 'lower', the [CHANGE] relationship was replaced with: 'a decrease in'. When the [RELATIONSHIP] variable was left empty, the [CHANGE] relationship was left empty, recreating the structure of Study 1.

## Study 3

In Study 3, we gave participants a tertiary response rather than a binary response, allowing them to express that neither of the two entities were causally related. More specifically, the study replicated Study 2, condition A ('nonsense names'), with an additional response option: 'A third factor is causally related to [X] and [Y], they are not directly related', where X and Y were the same nonsense terms used in Study 2.

**Participants.**   We recruited 100 participants, matching the sample sizes of the different conditions in Studies 1 and 3. Written/verbal informed consent was obtained from all participants for inclusion in the study. Participants were US-based No participants were excluded. The mean age of participants was 34.8, 63 identified as female, and 37 identified as male.

**Stimuli.**   Study 3 following the logic and stimuli of Study 2A. Participants were informed that they would be asked a few simple questions about the possible relationship between different things, and that the questions were unrelated to one another. Same as Study 2, the ordering of the nonsense term pairs within each question, the ordering of the forced choice answers to each question, and the question order of the five questions was randomized.

## Data quality assurance

We appreciate the growing concern about using AI tools to answer online surveys, and we do not think there is an agreed-on safeguard yet against the use of latest tools like ChatGPT-4 to answer online catch questions. We would note that the first two studies were run in early 2022, when the use of large language models was not widespread, and their performance when used

was rather limited. Our catch question was meant to weed out low-effort, automatic responses such as 'have a good day or 'relationships'. To the degree that we did not weed out all auto-replies, we believe that they only introduced a bit of noise, and given that our studies report differential effects this extra noise on top of the main effect is not a primary concern. The third study was run in late 2022, around the advent of the (now legacy) version of ChatGPT, and before adjusting for how well (or not) such tools can pass various tasks. We cannot guarantee our last question weeds out malicious users of the latest automatic tools, and moving forward we plan to adjust our data validation. However, we believe that the timing of the study was such that it is unlikely a large amount of malicious users on Prolific (to the degree such a group existed) had time to mass adopt ChatGPT into their workflow and rack up hundreds of completed and approved studies in time for our study. We also note that the marginal financial gain of using these tools is likely outweighed by the cost of per-token expenses.

## Acknowledgments

We are grateful to Michael Franke for helpful discussions.

## Author Contributions

**Conceptualization:** Samuel J. Gershman, Tomer D. Ullman.

**Data curation:** Tomer D. Ullman.

**Formal analysis:** Tomer D. Ullman.

**Investigation:** Tomer D. Ullman.

**Visualization:** Tomer D. Ullman.

**Writing – original draft:** Samuel J. Gershman, Tomer D. Ullman.

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
