## [Decision Letter · Decision Letter 0]

19 Sep 2022

PONE-D-22-19317Causal implicatures from correlational statementsPLOS ONE

Dear Dr. Gershman,

Thank you for submitting your manuscript to PLOS ONE. After careful consideration, we feel that it has merit but does not fully meet PLOS ONE’s publication criteria as it currently stands. Therefore, we invite you to submit a revised version of the manuscript that addresses the points raised during the review process.Reviewer 1 was more positive than Reviewer 2. R1 carefully spells out multiple possible interpretations of the results and suggests ways for the manuscript to account for these interpretations. R2 does not think the paper does an adequate job of engaging with the existing literature on this topic. The paper as it stands is quite short, so more adequately addressing the prior research could still be done in a "brief report" length paper. 

We look forward to receiving your revised manuscript.

Kind regards,

Micah B. Goldwater, Ph.D

Academic Editor

PLOS ONE

Journal Requirements:

"This work was supported by the Center for Brains, Minds and Machines (CBMM), funded by NSF STC award CCF1231216."

Reviewers' comments:

Reviewer's Responses to Questions

**Comments to the Author**

1. Is the manuscript technically sound, and do the data support the conclusions?

Reviewer #1: Partly

Reviewer #2: Partly

2. Has the statistical analysis been performed appropriately and rigorously? 

Reviewer #1: Yes

Reviewer #2: Yes

3. Have the authors made all data underlying the findings in their manuscript fully available?

Reviewer #1: Yes

Reviewer #2: Yes

4. Is the manuscript presented in an intelligible fashion and written in standard English?

Reviewer #1: Yes

Reviewer #2: No

5. Review Comments to the Author

Reviewer #1: This paper examines how people interpret statements of association (e.g. "X is associated with increased risk of Y"). Participants were asked to respond to statements like this in a forced-choice task, choosing whether it is more likely that "X causes Y" or "Y causes X." The authors find that participants are biased to interpret statements of association as implying specific causal directionality. They conclude that this is due to causal implicatures conveyed by the association statements, which the authors call the "causal implicature hypothesis."

Overall, the paper is nicely and clearly written. The methods appear appropriately sound and the results are very clear with respect to the statistical hypotheses tested. I also commend the open materials and data though I confess I have not examined the data myself.

I have a few points to raise for revisions:

First, a question about one condition in Study 2 and its relation to Study 1. In Figure 1, the "association" condition in S2 appears to show chance responding. From the methods, I understand this condition was essentially the same as Study 1, which showed non-chance responding. Am I correct in interpreting this as a failure to replicate S1's results in S2? If so, I think it would be best for the authors to address this---S1 is preregistered and the effects observed are sizable enough I would expect them to replicate, so it seems a bit puzzling.

Next, I have two other concerns: one I feel is essential to any revision and another I think would help to improve the quality of the discussion.

First, I have a major concern about the correspondence between some of the claims made in the introduction and discussion and what the results actually seem to support. For instance, in the abstract, the authors write "We show that people do in fact infer causality from statements of association, under minimal conditions." But the conditions under which participants inferred causality were forced choice tasks requiring them to infer causality, asking only about the most likely direction. So it does not seem justified to claim that people "infer causality" on the basis of these findings. In the discussion, the authors write "... participants inferred a causal relation, such that the first variable causes the second." Again it does not seem justified to foreground the first part of that claim in this way. In both cases it would be better to hew closer to what was actually shown. Maybe something like "participants were biased to infer specific causal directionality," making clear that they did not spontaneously make causal inferences. I do recognize that the authors address this somewhat in the discussion, but the claims should be made consistent with the findings throughout.

Second, I think more could be done to clarify conceptually what is being claimed and to distinguish it from other (potentially deflationary?) explanations of the findings. The authors aim to test what they call their "causal implicature hypothesis." The implicature hypothesis is that "there are implicit causal expectations about linguistic structures that are activated by correlational statements, and these expectations can be made explicit when people are forced to choose between different causal interpretations".

As I understand it, the idea is that statistical correlation or "association" is a symmetric or non-directional relationship. So, if people's interpretations of these statements are really committed to this symmetry, then they should choose randomly in the forced choice causal-direction tasks ("If people truly are committed to non-causal interpretations of non-causal language, then participants should just choose arbitrarily between the different causal interpretations available to them.") Instead, the authors' studies have shown that, when pressed, people interpret such statements as implying a specific causal directionality. The logic of these studies is that observing symmetric statements producing asymmetric interpretations gives evidence for the causal implicature hypothesis.

However, I see a few ways that the statements might not be seen as truly symmetric, especially in Study 2. Consider "increases risk of" statements: On the one hand it's true that if P(A|B) > P(A), then P(B|A) > P(B), but on the other hand the degree of the inequality might not be at all symmetric. For instance, some rare event A might greatly increase risk of B, yet A could still remain fairly unlikely given B. In such a case it would be literally true to say both that "A increases risk of B" and "B increases risk of A", but it might be pragmatically more informative to say "A increases risk of B". (I suppose whether this is so would depend on how people think about quantifying risk differences. In terms of mutual information things are symmetric, but in terms of differences like P(A|B) - P(A) they need not be.) So statements of association can sometimes be asymmetric without any causal content. Participants might then be aligning their causal directionality with the "directionality" of the original statements.

Second, the effects might not be due to implicatures but rather something like vagueness. One possibility is people might more directly interpret the claims about association as being based on experimental evidence. They might interpret the statement "X is associated with increased risk of Y" as "_doing_ X is associated with an increased risk of Y". To be sure, pragmatically it is odd for a speaker to say "associated with an increased risk of" rather than "increases risk" or "causes" in such a case. But, participants are ultimately given a forced choice between two causal interpretations. When forced to pick a causal interpretation, it seems like the statements in Study 2 at least leave open the "doing" reading, and hence would support the causal interpretations participants made. (note also that these "doing" interpretations are asymmetric.)

To address this concern I'd like to see more careful discussion of how the authors think participants actually interpret the statements in their studies and how they might interpret such statements outside the lab. I could be convinced that the objections I've raised are not major problems for what I take to be the spirit of the authors' conclusions: e.g. as I said I think you could construe part of my complaint as a question about whether their results are due to pragmatic implicatures or simply vagueness. In practice, we might expect similar consequences and thus it seems reasonable to care about both: Presented with careful statements of association by scientists, lay people will (at least sometimes) interpret them as loose statements about causal relationships. And on the other front, if statements of association have inherent (pragmatic) asymmetry, which lead to causal implications, that seems worth knowing too. Still I'd encourage the authors to clarify their claims and contrast their causal implicature hypothesis with alternate accounts. I'd also encourage them to consider how different accounts of their findings then relate to real-world consequences (e.g. especially in relation to their forced-choice task---without these pressures misinterpretations might be less likely).

Reviewer #2: Correlation does not imply causation and people are often warned of this fact. This suggests people may tend to ascribe causes even when purely correlational statements are made. This is the premise of the paper.

The paper is very brief and I'll keep my comments on it brief as well. Although the studies seem like reasonable first passes at studying the question, there is next to no discussion of the rich literature that exists and how different syntactic structures may prompt people to think an antecedent caused a consequent. Some of this work is cited. None of the details of these results are discussed to contextualize the studies that have been run.

In my view, this paper reads like a blog post---and that has some virtues--- but doesn't give enough to the reader enough to understand their findings, nor really engages with the fact that people have thought about this and related questions before. For the work to be publishable, the authors will need to write a paper instead of a blog post.

6. PLOS authors have the option to publish the peer review history of their article (what does this mean?). If published, this will include your full peer review and any attached files.

Reviewer #1: **Yes: **Derek Powell

Reviewer #2: No

---

## [Editor Report · Decision Letter 1]

2 May 2023

PONE-D-22-19317R1Causal implicatures from correlational statementsPLOS ONE

Dear Dr. Gershman,

Thank you for submitting your manuscript to PLOS ONE. We apologize for delay. After initially indicating they could review a revision, the original reviewers had to withdraw, and so to not delay further, I alone evaluated the manuscript. I think you adequately responded to the reviewers comments, and the paper presents interesting findings. I briefly add my own thoughts.I agree with your causal implicature account. It seems people hold a default assumption is that things happen because of causes or "for a reason" and there is work from multiple areas of cognition that people do not like having empty causal models (e.g., from the continued influence effect). If language is a consistent with a cause, then it will be interpreted as such if nothing else stops someone from doing so. There's a reason there needs to be explicit reminders that a correlation can always means a hidden third variable is the true cause of both A and B. Without that reminder, which at least suggests an alternative causal model, the only possible cause to infer is between A and B, and so it seems people do. That said, it's even unclear how successful the common-cause warnings are when they don't include a candidate third cause (or even when they do). Regardless of addressing these questions specifically, I'll be interested to see where this line of research is going next.  There is just a single minor revision needed before the paper can be accepted. There is a journal policy (that I was recently reminded of) that there needs to be safeguards of data quality for online data sources such as from Prolific, which you used here. Given the rise of bots and AI tools to answer surveys, this has become even more critical in  just the past six months. I see that  "participants were restricted to those located in the USA, and having completed at least 100 prior studies on Prolific, with an acceptance rate of at least 90%." And further, they were asked "Please describe, in a few words, what you were asked to do in this experiment?”" at the end of the study.  If there is further evidence to provide that when people automate responses, they are likely to be rejected (and so would not reach a 90% acceptance rate), and whether this last sentence would be difficult to automatically generate a response that actually reflected the content of the study, that would help increase the confidence in the data quality.  I assume providing this further evidence will not be much of a burden. 

We look forward to receiving your revised manuscript.

Kind regards,

Micah B. Goldwater, Ph.D

Academic Editor

PLOS ONE
---

## [Editor Report · Decision Letter 2]

9 May 2023

Causal implicatures from correlational statements

PONE-D-22-19317R2

Dear Dr. Gershman,

We’re pleased to inform you that your manuscript has been judged scientifically suitable for publication and will be formally accepted for publication once it meets all outstanding technical requirements.

Kind regards,

Micah B. Goldwater, Ph.D

Academic Editor

PLOS ONE
---

## [Editor Report · Acceptance letter]

10 May 2023

PONE-D-22-19317R2 

Causal implicatures from correlational statements 

Dear Dr. Gershman:

I'm pleased to inform you that your manuscript has been deemed suitable for publication in PLOS ONE. Congratulations! Your manuscript is now with our production department. 

Kind regards, 

on behalf of

Dr. Micah B. Goldwater 

Academic Editor

PLOS ONE